# Diversity of Bacteria with Quorum Sensing and Quenching Activities from Hydrothermal Vents in the Okinawa Trough

**DOI:** 10.3390/microorganisms11030748

**Published:** 2023-03-14

**Authors:** Fu Yin, Di Gao, Li Yue, Yunhui Zhang, Jiwen Liu, Xiao-Hua Zhang, Min Yu

**Affiliations:** 1Frontiers Science Center for Deep Ocean Multispheres and Earth System, College of Marine Life Sciences, Ocean University of China, 5 Yushan Road, Qingdao 266003, China; 2Laboratory for Marine Ecology and Environmental Science, Laoshan Laboratory, Qingdao 266237, China; 3Institute of Evolution & Marine Biodiversity, Ocean University of China, Qingdao 266003, China

**Keywords:** quorum sensing, quorum quenching, bacterial diversity, hydrothermal vents

## Abstract

Quorum sensing (QS) is a chemical communication system by which bacteria coordinate gene expression and social behaviors. Quorum quenching (QQ) refers to processes of inhibiting the QS pathway. Deep-sea hydrothermal vents are extreme marine environments, where abundant and diverse microbial communities live. However, the nature of chemical communication in bacteria inhabiting the hydrothermal vent is poorly understood. In this study, the QS and QQ activities with *N*-acyl homoserine lactones (AHLs) as the autoinducer were detected in bacteria isolated from hydrothermal vents in the Okinawa Trough. A total of 18 and 108 isolates possessed AHL-producing and AHL-degrading abilities, respectively. Bacteria mainly affiliated with *Rhodobacterales*, *Hyphomicrobiales*, *Enterobacterales* and *Sphingomonadales* showed QS activities; QQ was mainly associated with *Bacillales*, *Rhodospirillales* and *Sphingomonadales*. The results showed that the bacterial QS and QQ processes are prevalent in hydrothermal environments in the Okinawa Trough. Furthermore, QS significantly affected the activities of extracellular enzymes represented by β-glucosidase, aminopeptidase and phosphatase in the four isolates with higher QS activities. Our results increase the current knowledge of the diversity of QS and QQ bacteria in extreme marine environments and shed light on the interspecific relationships to better investigate their dynamics and ecological roles in biogeochemical cycling.

## 1. Introduction

Quorum sensing (QS) is a bacterial communication system that enables bacterial populations to regulate their group behaviors according to density [1]. QS relies on the production, diffusion, accumulation and group-wide detection of extracellular signaling molecules termed autoinducers [2]. QS coordinates various intra- and inter-bacterial behaviors, such as extracellular enzyme production [3], DNA uptake [4] and biofilm formation [5]. It has been estimated that a significant portion of the bacterial genome (4–10%) and proteome (≥20%) is influenced by QS, suggesting that QS is important for the adaptation of bacteria to various environments [6]. A range of autoinducers has been discovered and identified. Thus, QS mediated by *N*-acyl homoserine lactones (AHLs) has been widely studied in a diverse range of Gram-negative bacteria. Moreover, the AHL-producing genes have been identified in many bacteria, including *luxI* which leads to the formation of 3OC_6_-HSL in *Vibrio* spp., and *lasI* which is involved in 3OC_10_-HSL in *Pseudomonas aeruginosa* [7].

Marine bacteria with AHL-mediated QS (AHL-QS) activities have been isolated from multiple marine habitats, such as marine snow [8], corals [9,10], sponges [11], dinoflagellates [12], *Trichoderma* colonies [13] and sea anemones [14]. These bacteria belong mainly to *Alphaproteobacteria* and *Gammaproteobacteria* [15]. In addition, previous studies have shown that *Actinobacteria* [3,16], *Firmicutes* [16,17], archaea [18] and cyanobacteria [19] could produce AHLs. Doberva et al. [20] identified a large number of *luxI* genes in the global ocean metagenomes, indicating that AHL-QS may be of great significance for communication among microorganisms, and the regulation of biological behaviors in marine habitats. Additionally, AHL-QS was associated with large-scale bioluminescence events caused by algal blooms [21] and was important in maintaining the health of the coral reef ecosystem [22].

AHL-QS exerts an important role in regulating the elements cycling in the global ocean by affecting the activity of extracellular hydrolase and biofilm formation. For example, Jatt et al. [8] determined that QS could regulate bacteria to attach to marine snow particles and produce exoenzymes, thereby affecting the degradation of marine particulate organic carbon. Moreover, the biofilm formation and lipase production of *Ruegeria mobilis* Rm01, which was isolated from marine particles, could be regulated by QS [3]. In addition, adding AHL molecules to the *Trichodesmium* consortia could double the activity of phosphatase in the epibiotic bacteria, thereby promoting phosphorus circulation [13].

Quorum quenching (QQ) inhibits QS pathways by interfering with the production, release, recognition, and degradation of signaling molecules, thereby affecting various physiological functions dependent on bacterial QS [23,24]. Indeed, bacterial signaling molecules in the microenvironment could be removed by QQ activity [25], which was a crucial process for bacteria to maintain environmental homeostasis [2]. Bacteria with QQ activities have been isolated from various marine habitats, including biofilms [26], estuaries [23], corals [9] and sediment [27]. In addition, it was reported that QQ enzymes could cause decreases in autoinducer release and the activities of extracellular pectolytic enzymes to attenuate pathogenicity [28]. Despite their important ecological implications, the function of QS and QQ, in deep-sea hydrothermal vents have been poorly characterized [2].

In deep-sea hydrothermal vents, the dramatic changes in temperature (approximately 4–301 °C) and pH (ca. 5.0–7.6) vary in different marine environments [29]. Compared with other hydrothermal vents, the vent fluids from hydrothermal systems of the Okinawa Trough, influenced by the sediment from Chinese marginal seas, have lower pH and higher concentrations of CO_2_, CH_4_, NH_4_^+^, iodine and potassium [29]. In addition, the unique physicochemical conditions associated with hydrothermal vent systems give rise to abundant microbial communities. For example, there are abundant magnetotactic bacteria, sulfur-oxidizing bacteria (*Sulfurimonas* and *Sulfurovum*), sulfate-reducing bacteria (*Thermosulfidibacter* and *Desulfothermus*) and methanogens (*Methanococcales* and *Methanosarcinales*) living in the hydrothermal vents [30,31,32,33,34,35,36]. However, the interspecies relationship of microbial communities and their ecological function in hydrothermal vents from Okinawa Trough has been rarely studied.

In this study, we explored the diversity of QS and QQ isolates, and the regulatory function of QS on extracellular enzymatic activities in bacteria isolated from hydrothermal fields of the Okinawa Trough. The AHL-based QS and QQ activities of culturable bacteria were screened, and the biofilm formation of QS and partial-QQ strains was detected to reveal the relationship between QS/QQ and biofilm production. In addition, the effects of QS on the extracellular enzymatic activities of QS bacteria were explored.

## 2. Materials and Methods

### 2.1. Isolation and Identification of Bacterial Strains from the Okinawa Trough

Bacteria used in this study were isolated from seawater, sediment and macroorganisms (mussel, crab and shrimp) from the hydrothermal fields of the Okinawa Trough during the HOBAB2 voyage in 2014 and the HOBAB4 voyage in 2016 and 2018. The samples were collected by a TV-grab sampler or the remotely operated vehicle (ROV) Faxian during the cruises conducted by the R/V Kexue. The information about samples, including longitude and latitude and depth, were listed in Appendix A).

The seawater samples were serially diluted from 10^−1^ to 10^−3^, and the sediment samples were soaked in 0.85 % (*w/v*) saline serially diluted from 10^−1^ to 10^−3^, and 100 μL of each dilution was spread onto the surface of marine agar 2216E (MA) plates [37] and R2A plates [38]. The macroorganisms were dissected aseptically, and the tissues, including gill and intestine, were washed with 100 mL of 0.22 µm filtered seawater, which was then diluted and spread onto MA and R2A plates. The plates were cultured at 28 °C, and colonies were purified by streaking and re-streaking three to four times on MA or R2A plates. Further identification of the isolates relied on 16S rRNA gene amplification using the primers B8F (5′-AGAGTTTGATCCTGGCTCAG-3′) and B1510R (5′-GGTTACCTTGTTACGACTT-3′) followed by sequencing at Sangon Biotech (Qingdao, China). Sequence similarities between isolates and their most closely related bacteria were calculated using the EzBiocloud server (accessed on 20 July 2022. https://www.ezbiocloud.net). The phylogenetic tree of 16S rRNA genes from QS strains, their closest type strains and the reported AHL-producing strains was conducted using MEGA v11 based on the neighbor-joining algorithm [39]. The phylogenetic tree of QQ strains was based on their 16S rRNA gene sequences using MEGA v11 based on the neighbor-joining algorithm [39].

### 2.2. Screening for AHL-Producing and AHL-Degrading Bacteria

Screening for AHL-producing and AHL-degrading bacteria was conducted using the bacterial biosensor *Agrobacterium tumefaciens* (pCF218) (pCF372) A136 [40,41,42]. AHL-producing strains were identified by the co-feeding method according to Chu et al. [43]. Briefly, the bacterial cultures were inoculated into marine broth 2216E (MB) and cultured with shaking (170 rpm) at 28 °C for 24 h. *A*. *tumefaciens* A136 was added into the LB medium with 1% agar and X-gal (250 μg/L), and then the Oxford cups were placed on the plate to form wells. Meanwhile, fresh cultures were added into the wells, and the C_6_-HSL and MB were used as positive and negative controls, respectively. After co-incubation at 28 °C for 24 h, positivity was indicated by the presence of indigo spots. In addition, the high-throughput method described by Su et al. [3] was used to confirm the QS activities of these cultures. The fresh culture and an A136 X-gal assay solution [an overnight broth culture of A136 inoculated in AT minimal glucose medium [44] and mixed with X-Gal (250 μg/L)] were mixed and added into the 96-well microtiter plates, and the results were observed after cultivation at 28 °C for 24 h. C_6_-HSL and MB were used as positive and negative controls, respectively. All the experiments were performed in quadruplicate.

AHL-degrading activity was detected using the high-throughput method described by Tang et al. [45]. C_6_-HSL (representative for AHLs with short acyl chains) and C_12_-HSL (representative for AHLs with long acyl chains) were used as the substrates. Briefly, C_6_-HSL or C_12_-HSL was mixed with the bacterial culture and MB medium (negative control), and maintained at 28 °C for 24 h. After incubation, the supernatant was obtained after centrifugation at 4 °C and 6000 rpm for 10 min, and filtered through a 0.22 μm filter (JINTENG^®^ PES syringe filter). The supernatant was then mixed with an A136 X-gal assay solution in 96-well microtiter plates, and incubated at 28 °C for another 24 h. All the experiments were performed in triplicate. Positive results were indicated by reduced indigo color compared with the negative control.

### 2.3. Identification of QQ Activities in QS Strains

The QQ activities of QS strains were further identified using the high-throughput method. In short, QS strains were inoculated into MB with shaking (170 rpm) at 28 °C for 24 h. The cells were harvested after centrifugation at 4 °C and 6000 rpm for 10 min and re-suspended in HEPES buffer (20 mM Na-HEPES with 0.5 M NaCl, 10% glycerol, and 0.1% Triton X-100, pH 8.5) for sonication. Additional centrifugation for the lysed cells was conducted at 4 °C and 12,000 rpm to obtain the crude enzyme supernatant. The crude enzyme supernatant, C_6_-HSL or C_12_-HSL, HEPES buffer, and A136 X-gal assay solution were mixed and added into a 96-well microtiter plate, and incubated at 28 °C for 24 h. All the experiments were performed in triplicate. Positive results were indicated by a reduced indigo color compared with the negative control.

### 2.4. Biofilm Production in QS and QQ Strains

Biofilm production of all the bacteria with QS activities, and 23 isolates with C_6_- and C_12_-HSL-degrading abilities were quantified using a crystal violet (CV) assay, as described previously [25] with slight modifications. Briefly, the bacterial cultures were grown in MB, and 1% of the suspension was inoculated in MB medium contained in 96-well microtiter plates with incubation at 28 °C for 18 h. Each subsequent step was performed gently to preserve the biofilm. Thus, the supernatant was removed, and non-adherent cells were washed with sterile double distilled water. The 96-well microtiter plates were dried, and methanol was added to fix the cells; biofilms were stained with 0.2% (*w/v*) CV. Then, the CV in the biofilm was dissolved with 150 μL of 95% acetic acid, and the resulting OD was measured at 570 nm (OD570). MB was used as a negative control, and all the experiments were performed in triplicate. The significant difference between the experimental groups and the controls was calculated by the t-test (*** *p* < 0.001, ** *p* < 0.01, and * *p* < 0.05).

### 2.5. Extracellular Enzyme Activities in QS Strains

The extracellular enzyme (EE) activities were quantified following the methods described by Hmelo [46]. Five EE activities were tested using fluorescent substrates, including MUF-α-glucopyranoside (for α-glucosidase activity), MUF-β-glucopyranoside (for β-glucosidase activity), MUF-butyrate (for lipase activity), MCA-leucine (for aminopeptidase activity) and MUF-phosphate (for phosphatase activity) (all fluorescent substrates were purchased from Sigma-Aldrich). In brief, bacterial isolates were grown in MB at 28 °C, and the supernatants were retained after centrifugation at 4 °C and 12,000 rpm for 10 min and filtering through 0.22 μm filters. The supernatants and fluorescent substrates were mixed and added into a fluorescent multi-well plate with incubation at 28 °C for 12 h. Released fluorescent signals were detected with a Fluoroskan Ascent FL multi-well plate reader. The excitation and emission characteristics of the fluorophores were previously programmed into the instrument. In addition, the effects of QS on the EE activities in the isolates with higher QS activities were detected. The AHL lactonase MomL, which could degrade both short- and long-chain AHLs with or without substitution of oxo-group at the C-3 position [47], was used to test the effects of QS on the EE activities. Isolates were grown with purified MomL (1 U mL^−1^) [47] or inactive MomL (with boiling to inactive) in MB at 28 °C, 170 rpm for 48 h, and the supernatants were obtained after centrifugation at 4 °C and 12,000 rpm for 10 min at intervals of 12 h. MB with MomL and inactive MomL were used as controls. Released fluorescent signals were detected, as described above, and the EE activities were tested after the supernatants and fluorescent substrates were mixed and incubated at 28 °C for 3 h. All the experiments were performed in triplicate. The variations of EE activities were calculated by subtracting the EE activities of blank controls from that of the corresponding treatments. The significance of the difference between the treatments with and without MomL was calculated by the *t*-test (*** *p* < 0.001, ** *p* < 0.01, and * *p* < 0.05).

## 3. Results

### 3.1. AHL-Producing Bacteria Isolated from the Hydrothermal Fields of the Okinawa Trough

To reveal the diversity of QS bacteria isolated from deep-sea hydrothermal vents, the AHL-producing activities of bacteria isolated from hydrothermal fields in the Okinawa Trough were detected. A total of 305 isolates belonging to 178 species have been recovered from seawater, sediment and macroorganisms in hydrothermal fields of the Okinawa Trough in a previous study. After the preliminary selection, a total of 159 isolates belonging to 158 different species were chosen to examine their QS activities. There were 56, 55 and 48 strains isolated from sediment, seawater and macroorganisms, respectively (Appendix A). According to the 16S rRNA gene sequence analyses, these isolates belonged to *Proteobacteria*, *Bacteroidetes*, *Actinobacteria* and *Firmicutes* (Appendix A). Among them, 18 cultures showed AHL-producing activities, including eight isolates from macroorganisms (16.67%), seven from seawater (12.73%) and three from sediment (5.35%) (Table 1). These AHL-producing bacteria belonged to the orders *Rhodobacterales* (4), *Hyphomicrobiales* (3), *Enterobacterales* (3), *Sphingomonadales* (2), *Pseudomonadales* (1), *Cellulomonadales* (1), *Dermabacterales* (1), *Cytophagales* (1), *Flavobacteriales* (1) and *Bacillales* (1) (Table 1 and Figure 1). Four isolates exhibited relatively higher QS activities, and were equated with *Nitratireductor indicus* LLJ939, *Thalassococcus profundi* RWAS1, *Stakelama pacifica* LLJ869 and *Pseudohoeflea suaedae* SCR2 (Figure 2 and Appendix A).

### 3.2. Identification of Species Capable of AHL Degradation

The AHL-degrading activities of the 159 isolates were further examined, and C_6_-HSL and C_12_-HSL were used as the representative AHLs with short and long acyl chains, respectively. Among them, 60 (38.37%) and 89 cultures (55.98%) showed C_6_-HSL degradation and C_12_-HSL degradation, respectively. A total of 41 (25.78%) isolates were capable of degrading both C_6_-HSL and C_12_-HSL (Appendix A and Figure 3). Consequently, there were 108 isolates (67.92%) having the ability of degrading AHLs.

The 108 bacterial isolates with AHLs degrading activities belong to *Alphaproteobacteria* (35), *Gammaproteobacteria* (26), *Betaprotebacteria* (3), *Firmicutes* (32), *Actinobacteria* (8) and *Bacteroidetes* (4). At the order level, it was shown that the QQ strains were mainly affiliated with *Bacillales* (33), *Rhodospirillales* (19), and *Sphingomonadales* (9). Furthermore, bacteria showing both C_6_-HSL- and C_12_-HSL-degrading activities were mainly affiliated with *Bacillales* (10) and *Rhodobacterales* (9) (Table 2 and Figure 3).

For bacteria isolated from seawater samples, 20 (41.07% of isolates from seawater) and 29 (52.73%) cultures exhibited the C_6_- and C_12_-HSL-degrading activities, respectively. Among the isolates from sediment samples, 23 (41.07%) and 33 (58.93%) showed C_6_- and C_12_-HSL-degrading activities, respectively. As for cultures from macroorganism, 17 (35.41%) and 33 (56.25%) of the isolates were capable of degrading C_6_- and C_12_-HSL, respectively. The QQ cultures isolated from macroorganism sand seawater samples belonged mainly to *Alphaproteobacteria* and *Gammaproteobacteria*. QQ strains isolated from sediment were affiliated mainly with *Firmicutes.* It is noteworthy that more bacteria from sediment samples were found to exhibit QQ activities (Table 3 and Appendix A).

Among bacteria with QS activities, six cultures showed C_6_-HSL-degrading activities, including *Klebsiella michiganensis* BODM11, *Cellulomonas taurus* BOS2, *N*. *indicus* LLJ939, *Sphingobium yanoikuyae* RASR5, *Thalassococcus profundi* RWAS1 and *Yoonia rosea* YESM7. Meanwhile, five cultures exhibited C_12_-HSL-degrading activities, including *Arenibacter palladensis* CCM2, *Roseovarius indicus* CCR3, *Cyclobacterium marinum* CCS19, *N*. *indicus* LLJ939 and *S*. *yanoikuyae* RASR5. QS strains *N*. *indicus* LLJ939 and *S*. *yanoikuyae* RASR5 exhibited both C_6_-HSL- and C_12_-HSL-degrading activities (Figure 3).

### 3.3. Biofilm Production in QS and QQ Cultures

In order to preliminarily reveal the relationship between QS/QQ and biofilm production, the CV assay was used to detect the biofilm production of 18 and 23 isolates with QS activities and QQ activities, respectively. Among them, 11 QS cultures demonstrated the ability to produce biofilm, with significantly stronger capabilities found in *A. palladensis* CCM2, *S*. *yanoikuyae* RASR5 and *R*. *indicus* CCR3 (Figure 4A). A total of 22 QQ isolates exhibited the abilities of biofilm production with higher production in *Limimaricola variabilis* RASR1, *Pelagerythrobacter marinus* BOSR2, *Ruegeria conchae* YESR12, *Marinobacter salarius* SQM1, *Marinobacter algicola* RWAS6 and *Muricauda ruestringensis* CCR15 (Figure 4B).

### 3.4. Extracellular Enzyme Activities in QS Isolates

To reveal how the QS regulated the extracellular enzymes in QS isolates, MUF-α-glucopyranoside, MUF-β-glucopyranoside, MUF-butyrate, MCA-leucine and MUF-phosphate were selected as fluorescent substrates for the activities of α-glucosidase, β-glucosidase, lipase, aminopeptidase and phosphatase, respectively. The EE activities of the four cultures with higher QS activities, including *N*. *indicus* LLJ939, *T*. *profundi* RWAS1, *S*. *pacifica* LLJ869 and *P*. *suaedae* SCR2, were detected. It was found that neither of the four QS isolates showed α-glucosidase or lipase activity. However, *T*. *profundi* WRAS1 and *S*. *pacifica* LLJ869 exhibited β-glucosidase activities, *S*. *pacifica* LLJ869 and *P*. *suaedae* SCR2 revealed phosphatase activities, and *N*. *indicus* LLJ939 and *S*. *pacifica* LLJ869 had aminopeptidase activities. After AHL lactonase MomL was added into the cultures to degrade the AHLs, the activities of β-glucosidase in *T*. *profundi* WRAS1 (at 12, 24, 36 and 48 h; Figure 5A), the activities of aminopeptidase in *N*. *indicus* LLJ939 (at 48 h; Figure 5C) and *S*. *pacifica* LLJ869 (at 24, 36 and 48 h; Figure 5D), and the activities of phosphatase in *P*. *suaedae* SCR2 (at 48 h; Figure 5E) and *S*. *pacifica* LLJ869 (at 12, 24, 36 and 48 h; Figure 5F) were significantly lower than that in these strains with inactive MomL. Moreover, it was shown that the activities of β-glucosidase in *S*. *pacifica* LLJ869 at 12 and 24 h were lower than that in strain LLJ869 with inactive MomL, but higher at 36 h (Figure 5B). It was suggested that QS showed significant effects on the EE activities in these QS strains.

## 4. Discussion

Although the functions of QS and QQ have been extensively studied in marine environments, much less work has been conducted in the extreme environments, such as hydrothermal vents. Our study expanded the current knowledge of the diversity of QS and QQ isolates in hydrothermal fields, and the regulation of QS on bacterial behaviors. In this study, a total of 159 bacterial isolates, which were recovered from hydrothermal fields in the Okinawa Trough and belong to 158 different species, were selected to screen their QS and QQ activities. The diversity of QS and QQ bacteria in hydrothermal vents was revealed, and the relationship between QS and the EE activities was explored. Our results showed that abundant QS regulation existed in hydrothermal fields, which might play important roles in regulating bacterial metabolic pathways and even biogeochemical cycles in hydrothermal vents.

### 4.1. The Diversity of QS Strains in Hydrothermal Fields

Compared with marine snow, corals, sponges and marine particles, the hydrothermal vents have unique and complex environments, where abundant and various microbial populations live. However, the diversity of QS and QQ bacteria in hydrothermal fields has never been revealed before. In this study, bacterial isolates, which were recovered from hydrothermal fields, were selected to evaluate their AHL-based QS and QQ abilities using bacterial biosensor *A. tumefaciens* A136.

AHL synthase sequences belonging to *Alphaproteobacteria* have been reported in several environmental metagenomics datasets. Doberva et al. [20] reported that all sequences encoding for AHL synthases *luxI* in the Global Ocean Sampling dataset were derived from *Alphaproteobacteria* with 19 (65%), 3 (14%), 2 (7%) belonging to *Rhodobacterales, Rhizobiales* and *Sphingomonadales*, respectively. In addition, Su et al. [48] determined that *luxI* belonging to *Alphaproteobacteria* existed in particulate organic matter (POM) samples collected from coastal water and sediment in the Yellow Sea of China. In this study, 9 out of 55 isolates belonging to *Alphaproteobacteria* produced AHLs, with affiliation to *Rhodobacterales* (4), *Hyphomicrobiales* (3) and *Sphingomonadales* (2) (Figure 1); they were first reported to have QS activities at the species level. Many bacteria in the order *Rhodobacterales*, especially the *Roseobacter* Clade, have been found to have QS activities, which may play important roles in their association with marine invertebrates and nutrient-rich marine snows or organic particles [49]. For example, *Ruegeria* sp. KLH11 was isolated from a marine sponge, and could produce 3-OH-C_14_-HSL, 3-OH-C_12_-HSL and 3-OH-C_14_-HSL, which were engaged in temporal regulation of tropodithietic acid (TDA) production [50,51]. In our study, *R*. *indicus* CCR3, *Y*. *rosea* YESM7, *P*. *rhizosphaerae* CJG 283 and *T*. *profundi* WRAS1 were affiliated with the *Roseobacter* Clade and showed QS activities (Figure 1). Indeed, they may produce various AHLs and regulate their metabolic activities. Strains of *Sphingomonadales* from Yellow Sea particles [3] and Aulne estuary [52] have been found to possess QS activities. Similarly, *S*. *pacifica* LLJ869 and *S*. *yanoikuyae* RASR5 belong to *Sphingomonadales* and *S*. *pacifica* LLJ869 exhibited relatively higher QS activity, and the signaling molecules will be detected in the following study (Figure 1).

*Gammaproteobacteria* are among the most common AHL producers isolated from marine habitats [3,8]. However, in our study, only 4 out of 41 *Gammaproteobacteria* isolates were AHL producers. They were affiliated with *Enterobacterales* (3) and *Pseudomonadales* (1). *Enterobacterales* are usually found in marine aquatic environments and have emerged as major players in antimicrobial resistance worldwide. Jatt et al. [8] found that *Pantoea ananatis* B9, which belongs to *Enterobacterales* and was isolated from natural marine snow particles, could produce six AHLs, and its extracellular alkaline phosphatase activity was enhanced when adding exogenous AHLs. In our study, three isolates belonging to *Enterobacterales* showed QS activities, and they were all isolated from marine organisms inhabiting hydrothermal fields (Figure 1). It was suggested that there might be close relationships between bacteria and their hosts mediated by QS in hydrothermal vent environments [53].

*Actinobacteria* are widespread in marine environments and reached high abundance in deep-sea sediments, organic aggregates and macroalgae [54]. In terms of QS, several bacteria belonging to *Actinobacteria*, which were isolated from marine snow [3] and microbial mats in Shark Bay Australia [16] or reef macroalgae [55], have been identified to show AHL-producing activities. In the hydrothermal fields of the Okinawa Trough, we found that *C*. *taurus* BOS2 and *B*. *muris* LLJ752 affiliated with *Actinobacteria* showed AHL-QS activities (Figure 1). Although the QS activities of closely related bacteria in *Actinobacteria* were reported, our study provided the first report of AHL-producing bacteria isolated from hydrothermal vents in the Okinawa Trough. At present, a few studies have reported some AHL-producers belonging to *Firmicutes*, but the ecological importance and QS-regulated function is rarely studied [16,17]. In our study, *N*. *niacin* CJG092 belongs to *Firmicutes*, but it could not form biofilms. Therefore, the bacterial behaviors affected by QS in *Firmicutes* should be explored. In addition, few bacteria belonging to *Bacteroidetes* have been well studied in terms of QS [26]. In our study, two isolates belonging to *Bacteroidetes* showed QS activities, which may provide bacterial resources to find novel QS regulatory pathways in the future.

### 4.2. The Diversity of QQ Isolates in Hydrothermal Fields

At present, the metagenomic survey shows that genes encoding for both AHL acylase and lactonase showed relatively high abundance in marine environments [23]. Moreover, Romero et al. found that 14% and 18% of cultivable bacteria had been experimentally verified to harbor QQ activities in microbial biofilms and pelagic communities, respectively [23,56]. However, the diversity of QQ bacteria in hydrothermal fields has been rarely studied. In our study, 108 isolates (67.92%) showed QQ activities, and were mainly distributed in *Alphaproteobacteria*, *Gammaproteobacteria* and *Firmicutes*, which agreed with previous studies [2].

Many marine QQ bacteria belong to both *Alphaproteobacteria* [56], *Gammaproteobacteria* [57] and *Firmicutes* [58]. In our study, *Alphaproteobacteria* (64.81%), *Gammaproteobacteria* (63.41%) and *Firmicutes* (69.56%) showed a relatively higher proportion of QQ activities. Sequentially, Su et al. [3] found that *R*. *mobilis* Rm01 exhibited the degrading activities of C_10_-HSL, C_12_-HSL and C_14_-HSL because of AHL lactonase. Additionally, we found that both *Ruegeria atlantica* SIR11 and *Ruegeria arenilitorisi* CCM11, isolated from shrimp gut and crab gill, respectively, degraded C_12_-HSL, but only *R*. *atlantica* SIR11 attacked C_6_-HSL (Figure 3). Moreover, it was reported that *Pseudomonas sihuiensis* M4-84, which was isolated from the microbiota of sea anemones, had the capacity of degrading C_4_-HSL, C_6_-HSL, C_10_-HSL and C_12_-HS [59]. We found that *P*. *sihuiensis* CJG117 isolated from seawater exhibited QQ activity by degrading both C_6_-HSL and C_12_-HSL (Figure 3). Additionally, *P*. *sihuiensis* M4-84 was demonstrated to decrease the virulence of pathogens, and improve the survival rate of *Artemia salina* [59]. Similarly, we found that *B*. *cheonanensis* CJG088, *B*. *thioparans* RPSM8, *B*. *mycoides* RPZM19, *B*. *idriensis* TSR4, and *B*. *tianshenii* YESM4 could degrade both C_6_- and C_12_-HSL (Figure 3). Additionally, compared with short-chain AHLs, these QQ strains isolated from hydrothermal fields in the Okinawa Trough were more likely to degrade long-chain AHLs. It was reported that AHL-acylases preferred to degrade AHLs with longer carbon chain and could not degrade C_4_-HSL and C6-HSL in some bacteria [60,61]. Further, AHL acylases in cultivable marine bacteria [3,27,59,62] and in marine metagenomic collections [23,24] seems to be more abundant than AHL lactonases. In our study, more bacteria from sediment samples were found to exhibit QQ activities. It is possible that the abundance of bacteria was higher in sediments, and QQ might play more important roles in coordinating the relationships of various microorganisms.

In our study, 9 out of 18 QS strains exhibited degrading activities of AHLs; *N*. *indicus* LLJ939 and *S*. *yanoikuyae* RASR5 showed both C_6_-HSL- and C_12_-HSL-degrading activities (Figure 1). The reports of bacteria that simultaneously showed QS and QQ activities had been demonstrated before. For example, *R*. *mobilis* Rm01 not only produced 3OC_10_-HSL, C_10_-HSL and C_12_-HSL, but also had the capacity of degrading C_10_-HSL, C_12_-HSL and C_14_-HSL [3]. Strains belonging to *Marinobacterium* and *Shewanella* had the abilities of QS and QQ [63]. Furthermore, Urvoy et al. [52] found that ten QS strains showed QQ abilities to interfere with C_14_-HSL in the estuarine environment. QS and QQ pathways that co-existed were beneficial to the competition of microbial populations by limiting the growth and the coordination of bacteria taking part in QS communication [64,65], which could help maintain the homeostasis of the microbial community.

### 4.3. Biofilm Formation and the Regulation of EE Activities in QS Isolates

Biofilms may provide a relatively stable environment for bacteria to access resources, maintain the growth and reproduction of colonies, and protect the bacteria from detrimental conditions. It was reported that biofilm development processes were regulated by QS [66]. In our study, 11 QS isolates (61.11% of QS cultures) and 22 QQ isolates were able to produce biofilms (Figure 4 and Table 1). The capabilities of biofilm formation of QQ strains may be regulated by QS systems, other than AHL-QS, and the presence of biofilm had significant effects on whether bacteria could survive stably in extreme marine environments.

The QS pathways presented among marine bacteria may directly or indirectly influence biogeochemical cycles through the regulation of genes and EE activities, and the degradation of marine particles and marine snows [54]. Hmelo et al. [46] demonstrated that the addition of 3OC_6_-HSL or 3OC_8_-HSL improved the extracellular aminopeptidase and phosphatase activities of bacterial communities colonizing marine snow. Moreover, Krupke [67] showed that the addition of 3OC_8_-HSL significantly stimulated or inhibited hydrolytic phosphatase and aminopeptidase activities in sunken particle samples collected from the Atlantic and Pacific Oceans. Su et al. [3] reported that the addition of 3OC_8_-HSL to marine snow reduced the activities of β-glucosidase. In our study, we found that when the AHL-QS pathway was disrupted using MomL, the β-glucosidase, aminopeptidase and phosphatase activities were significantly decreased (Figure 5). The activities of β-glucosidase in *S*. *pacifica* LLJ869 with MomL were higher than the treatment with inactive MomL at 36 h, and it could be attributed to the drastic decrease in the β-glucosidase activities after the treatment of MomL after 24 h. It was shown that AHL-QS could affect the β-glucosidase, aminopeptidase and phosphatase activities in bacteria isolated from hydrothermal environments. Moreover, it is suggested that QS and QQ were essential, and complicated mechanisms engaged in microbial interactions and had important ecological implications in hydrothermal environments.

## 5. Conclusions

This study revealed the diversity of QS and QQ strains isolated from hydrothermal vents in the Okinawa Trough. The QS and QQ bacteria belong mainly to *Alphaproteobacteria*, *Gammaproteobacteria* and *Firmicutes*. The QS strains were mainly affiliated with *Rhodobacterales* (4), *Hyphomicrobiales* (3), *Enterobacterales* (3) and *Sphingomonadales* (2), and QQ strains mainly belong to *Bacillales* (33), *Rhodospirillales* (19) and *Sphingomonadales* (9). It implied the abundant QS and QQ microbial resources in the extreme marine environment. In addition, the diversity of QQ strains was higher than that of QS strains, and they were more likely to degrade long-chain AHLs compared with short-chain AHLs. The biofilm formation might help the QS and QQ strains survive in the hydrothermal environments. Moreover, the β-glucosidase, aminopeptidase and phosphatase activities in bacteria isolated from hydrothermal environments could be regulated by QS. In conclusion, our study revealed the diversity of culturable QS and QQ bacteria and explored the QS regulation of extracellular enzyme in deep-sea hydrothermal environments. In the future, the QS mechanisms and QQ enzymes will be further studied to explore the important roles of QS and QQ in regulating biogeochemical cycles in hydrothermal vents.

## Figures and Tables

**Figure 1 microorganisms-11-00748-f001:**
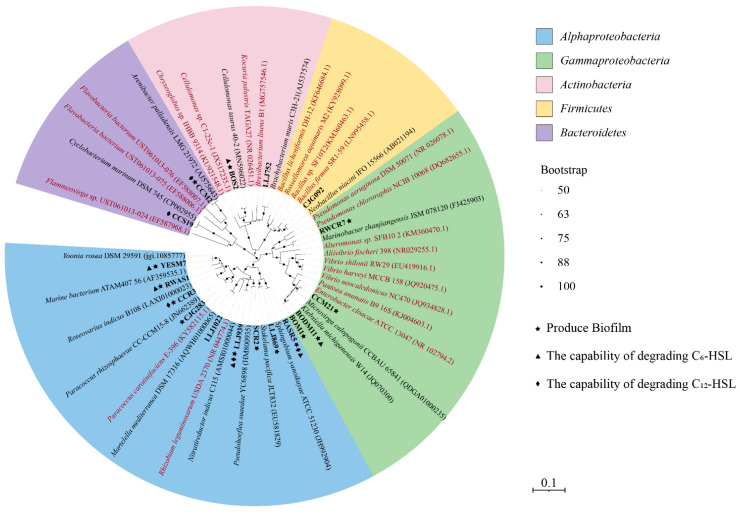
Phylogenetic tree of QS bacteria isolated from hydrothermal fields in the Okinawa Trough. The black blot at each branch point indicates the bootstrap values (>50%) based on a Neighbor-Joining analysis of 1000 resampled datasets. GenBank accession numbers of 16S rRNA gene sequences are given in parentheses. The characters in red indicate the QS strains found in the previous study, the characters in black and bold indicate the QS strains, the characters in black indicate the most similar strains of QS strains in this study.

**Figure 2 microorganisms-11-00748-f002:**
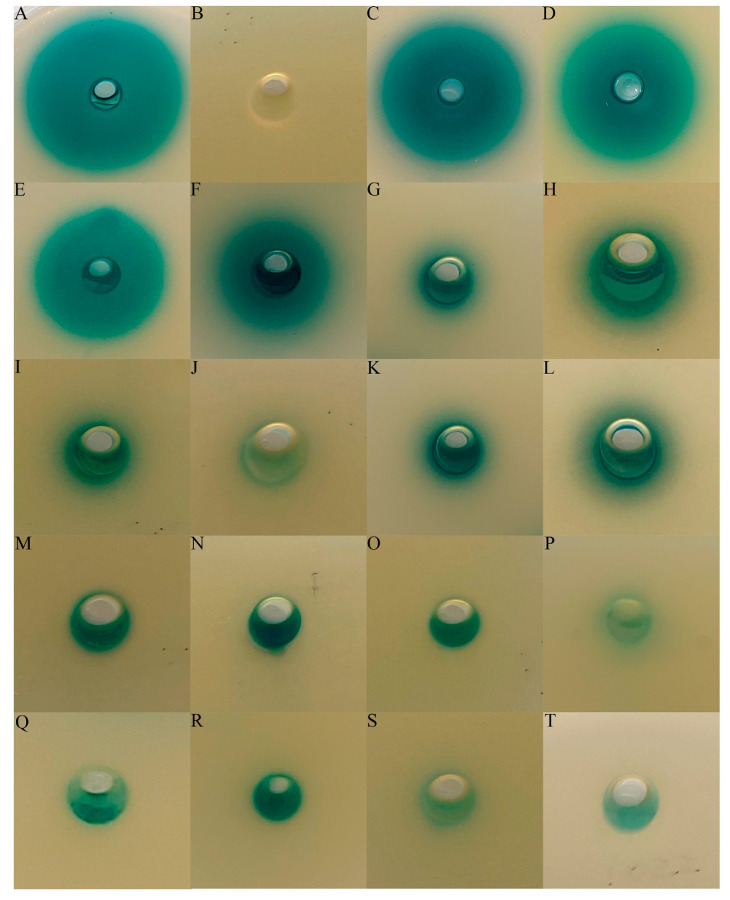
AHL-producing abilities of isolates recovered from hydrothermal fields in the Okinawa Trough. The results are shown via the co-feeding method supplemented with the AHL reporter strain *Agrobacterium tumefaciens* A136. (**A**,**B**), the positive control (C_6_-HSL, 10 nM) and the negative control (MB). (**C**–**T**) represent the following test strains, in order: *Nitratireductor indicus* LLJ939, *Thalassococcus profundi* RWAS1, *Stakelama pacifica* LLJ869, *Pseudohoeflea suaedae* SCR2, *Arenibacter palladensis* CCM2, *Yoonia rosea* YESM7, *Neobacillus niacini* CJG092, *Enterobacter hormaechei* BOM1, *Microvirga calopogonii* CCM21, *Cellulomonas taurus* BOS2, *Klebsiella michiganensis* BODM11, *Marinobacter zhanjiangensis* RWCR7, *Paracoccus rhizosphaerae* CJG283, *Cyclobacterium marinum* CCS19, *Brachybacterium muris* LLJ752, *Roseovarius indicus* CCR3, *Sphingobium yanoikuyae* RASR5, *Martelella mediterranea* LLJ1022.

**Figure 3 microorganisms-11-00748-f003:**
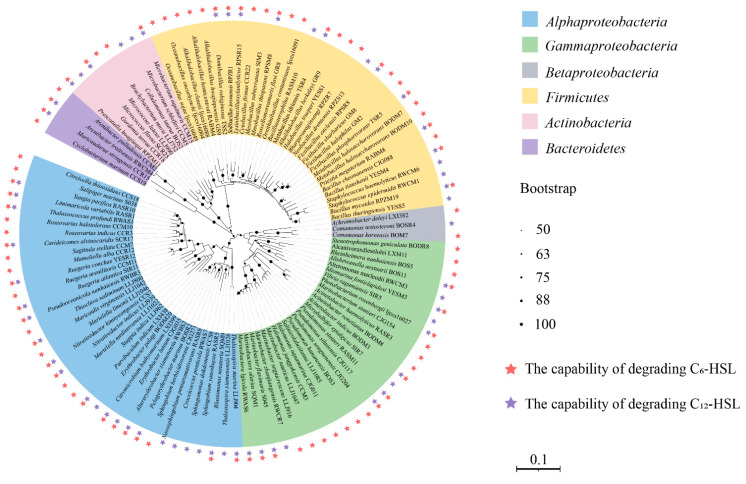
Phylogenetic tree of QQ bacteria isolated from hydrothermal fields in the Okinawa Trough. The black blot at each branch point indicates the bootstrap values (>50%) based on a Neighbor-Joining analysis of 1000 resampled datasets.

**Figure 4 microorganisms-11-00748-f004:**
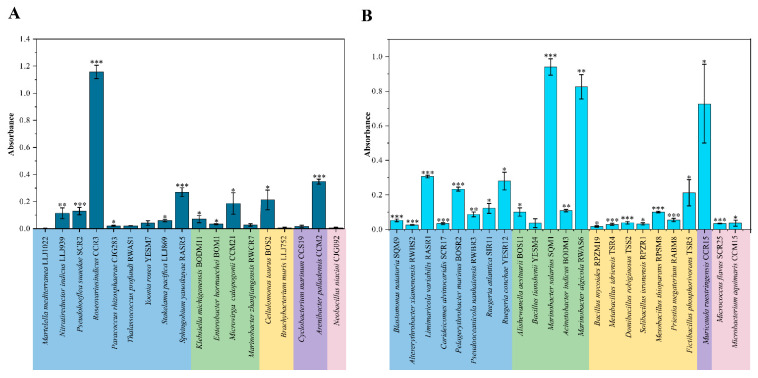
Biofilm production of bacterial isolates from the hydrothermal vent in the Okinawa Trough. (**A**), Biofilm production of QS strains. (**B**), Biofilm production of partial QQ strains. The data are shown as the mean ± SD, and the differences between the amended groups (the QS strains) and the control groups (the MB medium) are calculated by the t-test (*** *p* < 0.001, ** *p* < 0.01, and * *p* < 0.05).

**Figure 5 microorganisms-11-00748-f005:**
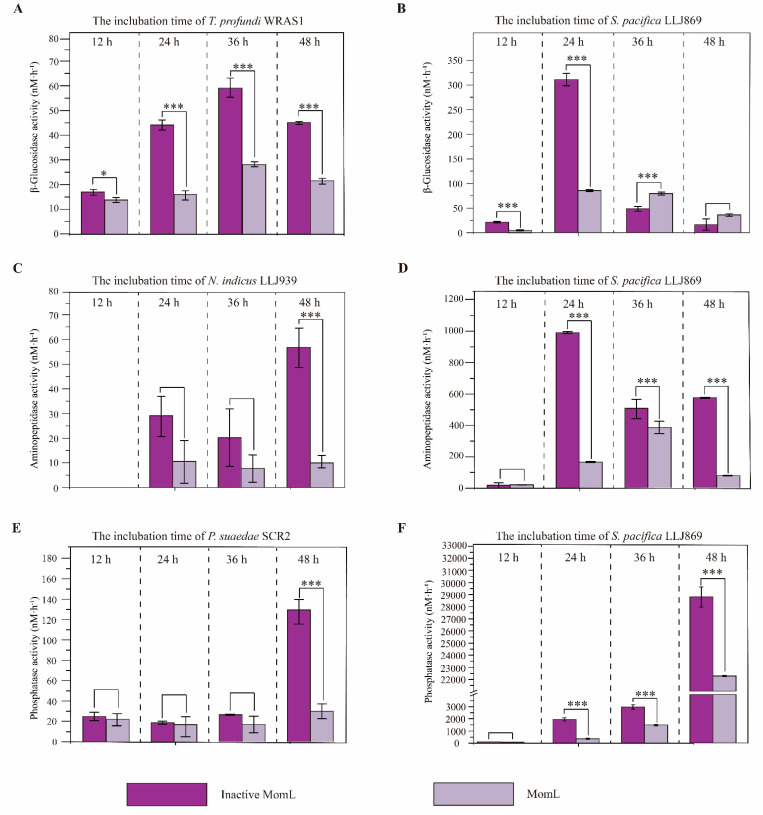
The β-glucosidase (**A**,**B**), aminopeptidase (**C**,**D**) and phosphatase activities (**E**,**F**) influenced by MomL. The data are shown as the mean ± standard deviation (SD). The differences between the amended groups (MomL) and the untreated groups (inactive MomL) are shown in different colors and calculated by the t-test (*** *p* < 0.001 and * *p* < 0.05).

**Table 1 microorganisms-11-00748-t001:** Information of QS strains isolated from hydrothermal fields in the Okinawa Trough.

Strain	Phylum	Class	Order	Species	Source *
LLJ1022	*Proteobacteria*	*Alphaproteobacteria*	*Hyphomicrobiales*	*Martelella mediterranea*	SD
LLJ939	*Proteobacteria*	*Alphaproteobacteria*	*Hyphomicrobiales*	*Nitratireductor indicus*	SW
SCR2	*Proteobacteria*	*Alphaproteobacteria*	*Hyphomicrobiales*	*Pseudohoeflea suaedae*	SG
CCR3	*Proteobacteria*	*Alphaproteobacteria*	*Rhodobacterales*	*Roseovarius indicus*	CG
CJG283	*Proteobacteria*	*Alphaproteobacteria*	*Rhodobacterales*	*Paracoccus rhizosphaerae*	SW
RWAS1	*Proteobacteria*	*Alphaproteobacteria*	*Rhodobacterales*	*Thalassococcus profundi*	SW
YESM7	*Proteobacteria*	*Alphaproteobacteria*	*Rhodobacterales*	*Yoonia rosea*	SD
LLJ869	*Proteobacteria*	*Alphaproteobacteria*	*Sphingomonadales*	*Stakelama pacifica*	SW
RASR5	*Proteobacteria*	*Alphaproteobacteria*	*Sphingomonadales*	*Sphingobium yanoikuyae*	SD
BODM11	*Proteobacteria*	*Gammaproteobacteria*	*Enterobacterales*	*Klebsiella michiganensis*	M
BOM1	*Proteobacteria*	*Gammaproteobacteria*	*Enterobacterales*	*Enterobacter hormaechei*	M
CCM21	*Proteobacteria*	*Gammaproteobacteria*	*Enterobacterales*	*Microvirga calopogonii*	CG
RWCR7	*Proteobacteria*	*Gammaproteobacteria*	*Pseudomonadales*	*Marinobacter zhanjiangensis*	SW
BOS2	*Actinobacteria*	*Actinomycetia*	*Cellulomonadales*	*Cellulomonas taurus*	M
LLJ752	*Actinobacteria*	*Actinomycetia*	*Dermabacterales*	*Brachybacterium muris*	SW
CCS19	*Bacteroidetes*	*Cytophagia*	*Cytophagales*	*Cyclobacterium marinum*	CG
CCM2	*Bacteroidetes*	*Flavobacteriia*	*Flavobacteriales*	*Arenibacter palladensis*	CG
CJG092	*Firmicutes*	*Bacilli*	*Bacillales*	*Neobacillus niacini*	SW

* SD, sediment; SW, seawater; SG, shrimp gill; CG, crab gill; M, mussel.

**Table 2 microorganisms-11-00748-t002:** The information of QQ strains isolated from hydrothermal fields in the Okinawa Trough (at the order level).

Phylum	Class	Order	The Number of Test Strains	The Number of Strains Degrading C_6_-HSL	The Number of Strains Degrading C_12_-HSL
*Proteobacteria*	*Alphaproteobacteria*	*Rhizobiales*	13	2	6
		*Rhodobacterales*	29	12	17
		*Sphingomonadales*	12	9	5
	*Gammaproteobacteria*	*Alteromonadales*	11	6	7
		*Pseudomonadales*	10	5	8
		*Enterobacterales*	4	0	0
		*Oceanospirillales*	9	0	5
		*Xanthomonadales*	1	0	1
		*Vibrionales*	7	2	2
	*Betaproteobacteria*	*Burkholderiales*	3	1	2
*Firmicutes*	*Bacillales*	*Bacillaceae*	46	15	28
*Actinobacteria*	*Actinomycetia*	*Corynebacteriales*	2	2	1
		*Micrococcales*	7	4	4
*Bacteroidetes*	*Flavobacteriales*	*Flavobacteriales*	4	2	3
	*Cytophagia*	*Cytophagales*	1	0	0
		Total number	159	60 (37.73%)	89 (55.97%)

**Table 3 microorganisms-11-00748-t003:** The numbers of QQ cultures isolated from hydrothermal fields in the Okinawa Trough.

Source	Total Number	The Number of Isolates Degrading C_6_-HSL (%)	The Number of Isolates Degrading C_12_-HSL (%)
Seawater	55	20 (36.36%)	29 (52.73%)
Sediment	56	23 (41.07%)	33 (58.93%)
Organism	48	17 (35.41%)	27 (56.25%)
Total number	159	60 (38.37%)	89 (55.98%)

## Data Availability

The data presented in this study are fully available in the main text and Appendix A of this article.

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
