# Peer review of "Diversity of Bacteria with Quorum Sensing and Quenching Activities from Hydrothermal Vents in the Okinawa Trough"

_microorganisms, 2023, doi:10.3390/microorganisms11030748_

Round 1
Reviewer 1 Report
Diversity of bacteria with quorum sensing and quenching activities from hydrothermal vents in the Okinawa Trough
Fu Yin , Di Gao , Li Yue , Yunhui Zhang, Jiwen Liu, Xiao-Hua Zhang and Min Yu
It is valuable to mention, that obtained results were original, innovative and demonstrated many goals associated with environmental microbiology and sustainable biotechnology. In this research, among 159 strains isolated, 18 and 108 strains showed AHL-producing and AHL-degrading abilities, respectively. Bacteria affiliated with Rhodobacterales, Hyphomicrobiales, Enterobacterales and Sphingomonadales showed QS (Quorum sensing) activities, and the QQ (Quorum quenching) bacteria mainly belonged to Bacillales, Rhodospirillales and Sphingomonadales. The results showed the bacterial QS and QQ processes were prevalent in hydrothermal environments in the Okinawa Trough. In addition, the effects of QS on the extracellular enzymatic activities of QS bacteria were explored. This study revealed the diversity of culturable QS and QQ bacteria and demonstrated the potential role of microbial QS and QQ in element cycling in deep-sea hydrothermal environments.
Some comments and suggestions to the authors:
1) I suppose, that should be written „Absorbance“ in the Y axis in the Figure 4.
2) To check writing of β-glucosidase activity dimension in the Y axis in the Figure 5A.
3) To check writing of phosphatase activity dimension in the Y axis in the Figure 5B.
4) To check writing of aminopeptidase activity dimension in the Y axis in the Figure 5C.
5) To check writing of aminopeptidase activity the dimension in the Y axis in the Figure 5D.
Author Response
Many thanks for the comments of the reviewer. We have carefully revised the manuscript in response to the reviewers. Changes in the revised manuscript are in red. Here are point-by-point responses:
1) I suppose, that should be written “Absorbance” in the Y axis in the Figure 4.
Reply: Thanks for your advice. We have revised them in Figure 4.
2) To check writing of β-glucosidase activity dimension in the Y axis in the Figure 5A.
Reply: Thanks. We have modified the β-glucosidase activity dimension.
3) To check writing of phosphatase activity dimension in the Y axis in the Figure 5B.
Reply: Thanks. We have modified the phosphatase activity dimension.
4) To check writing of aminopeptidase activity dimension in the Y axis in the Figure 5C.
Reply: Thanks. We have modified the aminopeptidase activity dimension.
5) To check writing of aminopeptidase activity the dimension in the Y axis in the Figure 5D
Reply: Thanks. We have modified the aminopeptidase activity dimension.
Reviewer 2 Report
This study describes a survey of quorum-sensing (QS) and quorum-quenching (QQ) bacteria collected from sediments, marine invertebrates and seawater in deep-sea hydrothermal vents. The authors identified a variety of isolates producing acyl-homoserine lactones (AHL), which represent one type of QS signals, in diverse phylogenetic bacterial groups. The species identified belong to groups found in marine environments and already known to have QS properties, but this is one of the first studies to analyze hydrothermal vents samples for QQ and QS. Many isolates displayed QQ activity, and the phylogenetic groups identified were in line with previous studies on marine samples.
This very descriptive study is not ground-breaking. Given the procedures used in this work it is very unlikely that new discoveries could be made. Only isolates that can grow in pure cultures were used, no analyses were conducted on selected isolates to identify the molecules produced or to define the sets of molecular functions regulated by QS in hydrothermal vent environments, and no QS molecules other than AHL were considered. Further studies will be necessary to reveal potential implications of QS systems in biogeochemical cycles. I also have concerns regarding the methods used to determine the enzymatic activities associated with QS.
Comments
- The link between QS and biofilm formation is tenuous, because the isolates producing largest amounts of AHL do not necessarily correspond to those with strong biofilm-forming properties. This should be discussed. In addition, some isolates with QQ but not QS properties should be included in the biofilm analyses as controls. It is possible that other QS systems exist in the isolates analyzed in this study and/or that biofilm formation occurs irrespective of AHL detection.
- What is the basis for the choice of the enzymatic activities tested other than the availability of fluorescent substrates ? Other enzymatic activities were described to be regulated by QS (proteases, lipases) but they were not searched for in this work. The authors should provide a rationale for their choice.
- It is unclear how the experiments that measure enzymatic activities in supernatants were performed, and if they were performed as stated, then they are not really meaningful. As I understand, supernatants from 24-h cultures were incubated with the substrates, and fluorescence readings were performed at regular intervals from 12 to 60 h to detect hydrolysis products. The timing for the addition of MomL is problematic, as it was added after supernatants were collected. At that stage the presence or absence of AHL makes no difference because there are no bacteria in the assay (but only supes) to respond to AHL by producing the extracellular enzymes. I thus do not understand the results shown in panel C of Fig 5, in which MomL addition abolishes enzymatic activity. If the assays were performed in a different manner (e.g., MomL was added to the 24-h cultures and not to the supernatants) then the authors need to revise their methods section.
- Even if MomL was added during the cultures, I do not see the point of measuring the fluorescence released beyond 12 h, as there is amply enough time for the enzyme to work during a 12-h period. The results shown at the other time points are difficult to make sense of.
- In Fig.S2 negative values of enzymatic activities are shown. The authors should clearly state that these do not mean anything.
- In one case (panel B of Fig 5) enzymatic activity slightly increased with the addition of lactonase. Provided that lactonase was added to the cultures and not to the supernatant (which is probably not the case; see my remark above) the explanation for this effect provided by the authors is not convincing. It is unclear how phosphatase activity could be negatively regulated by QS. Are there examples in the literature of negative regulation of extracellular enzymes by QS? If so, this should be discussed.
Minor comments
- The function of MomL should be mentioned in the Results section and not only in the discussion part. Is its expression regulated by QS in model species?
- I noted some awkward phrasing and incorrect grammar. The use of the tenses should be revised.
Author Response
Many thanks for the comments of the reviewer. We have carefully revised the manuscript in response to the reviewers. Changes in the revised manuscript are in red. Here are point-by-point responses:
1) The link between QS and biofilm formation is tenuous, because the isolates producing largest amounts of AHL do not necessarily correspond to those with strong biofilm-forming properties. This should be discussed. In addition, some isolates with QQ but not QS properties should be included in the biofilm analyses as controls. It is possible that other QS systems exist in the isolates analyzed in this study and/or that biofilm formation occurs irrespective of AHL detection.
Reply: Thanks for your advice, we have selected 23 QQ strains with C6-HSL and C12-HSL degrading abilities to test their biofilm production and added some discussion about the biofilm-forming properties found in QQ strains (lines 217-222 and lines 336-339).
2) What is the basis for the choice of the enzymatic activities tested other than the availability of fluorescent substrates? Other enzymatic activities were described to be regulated by QS (proteases, lipases) but they were not searched for in this work. The authors should provide a rationale for their choice.
Reply: Thanks. In our study, we have tested the lipases activities of the four strains, but it was found that neither of them showed lipases activities. We also have tested the activities of aminopeptidase, which could catalyze the cleavage of amino acids from the amino terminus of protein or peptide substrates. The activities of aminopeptidase in N. indicus LLJ939 and S. pacifica LLJ869 were regulated by AHL-QS (lines 228-230 and lines 234-239).
3) It is unclear how the experiments that measure enzymatic activities in supernatants were performed, and if they were performed as stated, then they are not really meaningful. As I understand, supernatants from 24-h cultures were incubated with the substrates, and fluorescence readings were performed at regular intervals from 12 to 60 h to detect hydrolysis products. The timing for the addition of MomL is problematic, as it was added after supernatants were collected. At that stage the presence or absence of AHL makes no difference because there are no bacteria in the assay (but only supes) to respond to AHL by producing the extracellular enzymes. I thus do not understand the results shown in panel C of Fig 5, in which MomL addition abolishes enzymatic activity. If the assays were performed in a different manner (e.g., MomL was added to the 24-h cultures and not to the supernatants) then the authors need to revise their methods section.
Reply: Many thanks for your advice. We have performed the experiments again according to your suggestions that the MomL was added into cultures and the EE activities were tested in 3 hours. We have modified the methods, results, and discussion parts (lines 141-155, lines 228-241 and lines 340-353).
4) Even if MomL was added during the cultures, I do not see the point of measuring the fluorescence released beyond 12 h, as there is amply enough time for the enzyme to work during a 12-h period. The results shown at the other time points are difficult to make sense of.
Reply: Many thanks for your advice. We performed the experiments again and abandoned the results after 12 hours. Based on the results before 12 hours without MomL, we have selected the strains with extracellular hydrolase activities to reveal how the QS regulated the activities of extracellular enzymes in QS strains (lines 141-155).
5) In Fig.S2 negative values of enzymatic activities are shown. The authors should clearly state that these do not mean anything.
Reply: Thanks. We have added the statement about the negative values in lines 38-40 of the supplementary materials.
6) In one case (panel B of Fig 5) enzymatic activity slightly increased with the addition of lactonase. Provided that lactonase was added to the cultures and not to the supernatant (which is probably not the case; see my remark above) the explanation for this effect provided by the authors is not convincing. It is unclear how phosphatase activity could be negatively regulated by QS. Are there examples in the literature of negative regulation of extracellular enzymes by QS? If so, this should be discussed.
Reply: Thanks. We have performed the experiments again according to your suggestions that the MomL was added into cultures. According to the new results, the phosphatase activity was positively regulated by QS (lines 234-239).
Minor comments
7) The function of MomL should be mentioned in the Results section and not only in the discussion part. Is its expression regulated by QS in model species?
Reply: Thanks. The AHL lactonase MomL, which could degrade both short- and long-chain AHLs with or without substitution of oxo-group at the C-3 position, was used to test the effects of QS on the EE activities. We have added the function of MomL in materials and methods (lines 147-148).
8) I noted some awkward phrasing and incorrect grammar. The use of the tenses should be revised.
Reply: Thanks. We have carefully checked the manuscript, and the awkward phrasing and incorrect grammar have been revised, and changes in the revised manuscript are in red.
Reviewer 3 Report
The subject studied by the authors of the manuscript is of a great interest. Communication between members of the microbial community is not well yet understood, therefore any results received in relation to this phenomenon are important.
The methodology used is well described, the results are clearly presented in tables and figures. The conclusions correspond to the tasks set.
Questions and suggestions are below:
1.Introduction. Lines 46-54. Are there data on archaea?
2. The manuscript presents the results of the study of the microbial community from deep-sea hydrothermal vents. It would be reasonable to give in the manuscript more information about the hydrothermal system under study (physical and chemical parameters (temperature, pH, chemical composition etc.), as well as its geographical localization. In the absence of such information, it looks like a common marine environment What parameters make this system very different (as authors indicate) from the normal marine environment?
3. P. 2, lines 67-69 and P. 12, lines 359-361. Both texts provide information on the ecosystems in which QQ bacteria are studied. Is this information similar? Should it be really repeated? If yes in what context?
4.P.2. line 74-75. “Although QS/QQ processes have been studied in diverse marine environments, the deep-sea hydrothermal vents own the complicated and changeable environments and the diversity of bacteria implying the potentially active processes of QS and QQ”
and also P.10. line 290-291. "Although QS and QQ activity and function have been extensively studied in marine environmental, much less work has been conducted on hydrothermal vents". As it is mentioned in the point (2), the complexity of deep-sea hydrothermal ecosystems and the diversity of bacteria in it should be additionally clarified. Why the diversity of microorganisms in hydrothermal vents is higher compared to other marine systems?
5.P.2. line 76-78. “The Okinawa Trough, which is a deep sea back-arc spreading basin located in the East China Sea, provides a unique setting to investigate the microbial community structure and function as well as the driven environmental factors on a regional scale [25].” This is a matter of style. What is the aforementioned uniqueness of the study? The uniqueness should be explained more clearly or changed with another term.
6.P.2. line 81-82. “The unique physicochemical conditions associated with different hydrothermal vent systems give rise to abundant microbial communities.” In addition to all above mentioned, it is necessary to give information about such unique physicochemical conditions.
7.P.2. Line 92-93. “Our study revealed the diversity of culturable QS and QQ bacteria and elucidated the potential role of microbial QS and QQ in element cycling in deep-sea hydrothermal environments”. It is better to transfer this sentence either to the Abstract or to the Conclusion.
8. Materials and Methods. Ii would be reasonable to add a paragraph about the description of the ecosystem in accordance with the above proposed corrections.
9.P.3. Line 97-114. It is necessary to describe in more detail the conditions for isolation of bacteria (temperature, pH, other parameters). Were the conditions just for moderate microorganisms in the extreme environment?
10.P.8. Line 247-248. “It is worth noting that more bacteria from sediment samples were found to exhibit the QQ activities (Table 3 & Table S1)”. Can you speculate on why?
11.P.13. Line 429-430. “In addition, the diversity of QQ strains was far more than that of QS strains, and they were more likely to degrade long-chain AHLs compared with short-chain AHLs.” Could you speculate on what is the reason for this fact?
Author Response
Many thanks for the comments of the reviewer. We have carefully revised the manuscript in response to the reviewers. Changes in the revised manuscript are in red. Here are point-by-point responses:
1.Introduction. Lines 46-54. Are there data on archaea?
Reply: Thanks for your advice. We have added some reports about archaea in lines 45-46.
- The manuscript presents the results of the study of the microbial community from deep-sea hydrothermal vents. It would be reasonable to give in the manuscript more information about the hydrothermal system under study (physical and chemical parameters (temperature, pH, chemical composition etc.), as well as its geographical localization. In the absence of such information, it looks like a common marine environment What parameters make this system very different (as authors indicate) from the normal marine environment?
Reply: Thanks. We have added more information about the hydrothermal systems in the introduction part (lines 65-69).
- P. 2, lines 67-69 and P. 12, lines 359-361. Both texts provide information on the ecosystems in which QQ bacteria are studied. Is this information similar? Should it be really repeated? If yes in what context?
Reply: Thanks. We agreed with you that the information is similar, and we have deleted the description in lines 311-312.
4.P.2. line 74-75. “Although QS/QQ processes have been studied in diverse marine environments, the deep-sea hydrothermal vents own the complicated and changeable environments and the diversity of bacteria implying the potentially active processes of QS and QQ” and also P.10. line 290-291. "Although QS and QQ activity and function have been extensively studied in marine environmental, much less work has been conducted on hydrothermal vents". As it is mentioned in the point (2), the complexity of deep-sea hydrothermal ecosystems and the diversity of bacteria in it should be additionally clarified. Why the diversity of microorganisms in hydrothermal vents is higher compared to other marine systems?
Reply: Thanks. The unique physicochemical conditions associated with different hydrothermal vent systems gave rise to abundant microbial communities. We have added more information to describe the physical and chemical parameters of the hydrothermal systems and the complexity and diversity of microorganisms in hydrothermal vents (lines 65-73).
5.P.2. line 76-78. “The Okinawa Trough, which is a deep sea back-arc spreading basin located in the East China Sea, provides a unique setting to investigate the microbial community structure and function as well as the driven environmental factors on a regional scale [25].” This is a matter of style. What is the aforementioned uniqueness of the study? The uniqueness should be explained more clearly or changed with another term.
Reply: Thanks. We have deleted the description and added new statements of the hydrothermal systems in the Okinawa Trough (lines 65-73).
6.P.2. line 81-82. “The unique physicochemical conditions associated with different hydrothermal vent systems give rise to abundant microbial communities.” In addition to all above mentioned, it is necessary to give information about such unique physicochemical conditions.
Reply: Thanks. We have added more information to describe the physical and chemical parameters of the hydrothermal systems and the complexity and diversity of microorganisms in hydrothermal vents (lines 65-73).
7.P.2. Line 92-93. “Our study revealed the diversity of culturable QS and QQ bacteria and elucidated the potential role of microbial QS and QQ in element cycling in deep-sea hydrothermal environments”. It is better to transfer this sentence either to the Abstract or to the Conclusion.
Reply: Thanks. We have modified and transferred this sentence to the Conclusion in lines 367-369.
- Materials and Methods. It would be reasonable to add a paragraph about the description of the ecosystem in accordance with the above proposed corrections.
Reply: Thanks. We have added a paragraph about the description of the hydrothermal ecosystem in lines 65-73.
9.P.3. Line 97-114. It is necessary to describe in more detail the conditions for isolation of bacteria (temperature, pH, other parameters). Were the conditions just for moderate microorganisms in the extreme environment?
Reply: Thanks. The bacterial isolates were cultured on MA or R2A plates or in MB at 28°C, and the conditions were for moderate microorganisms. We have added more information about the culture conditions (lines 90-91).
10.P.8. Line 247-248. “It is worth noting that more bacteria from sediment samples were found to exhibit the QQ activities (Table 3 & Table S1)”. Can you speculate on why?
Reply: Thanks. We supposed that the abundance of bacteria was higher in sediment, and QQ might play more important roles in coordinating the relationships of various microorganisms (lines 321-323).
11.P.13. Line 429-430. “In addition, the diversity of QQ strains was far more than that of QS strains, and they were more likely to degrade long-chain AHLs compared with short-chain AHLs.” Could you speculate on what is the reason for this fact?
Reply: Thanks. It was reported that AHL-acylases preferred to degrade AHLs with longer carbon chain and could not degrade C4-HSL and C6-HSL in some bacteria. Besides, it was also reported that AHL acylases in cultivable marine bacteria and in marine metagenomic collections seemed to be more abundant than AHL lactonases. We have added it in the discussion part (lines 317-321).
Round 2
Reviewer 2 Report
The authors have taken the advices provided to improve the results section. A major concern regarding the protocol of enzymatic activity measurements (i.e., the question of when the lactonase was added, to the cultures or to the supernatants) has been alleviated. This makes this set of data more convincing.
The work remains much of a catalog and is very descriptive, but this type of inventory is probably useful. I also noted some improvement in the discussion, although it mostly repeats the results section.
The authors have tried to revise the grammar, which nevertheless remains problematic. Because of this, the text is somewhat difficult to read.
Specific comments
Regarding the enzymatic activities tested, the authors should state that they also tested for lipase activity of the QS-proficient strains but did not detect any.
What is the point of Fig S2? The conditions in which the activities are measured in this figure are not explained, and therefore it is difficult to compare the data shown there with those in Fig. 5. Thus, the phosphatase activity of P. suaedae is 1600 nMh-1 in one case and 140 nMh-1 (at the maximum) in the other. In addition, even if negative values are said to mean nothing, they are present in the figure, which should not be the case. I suggest removing Fig S2. It is not useful.
Please have your manuscript checked by native English speakers. Problems with the tenses remain. In the introduction the present tense should be used for general statements. For instance, it should be:
Line 57: AHL-QS also plays an important role in regulating…
Lines 320-321: hydrothermal vents have unique…. microbial populations live.
This is also the case at many other places, which I decided was not my duty to point out.
Awkward/ incorrect sentences are found throughout the text. For instance:
Lines 352-354, it should be: …Pantoea ananatis B9, which belongs to Enterobacterales and was isolated from natural marine snow particles, could produce six AHLs, and its extracellular alkaline phosphatase activity …
Line 356, do the authors mean : It was suggested that … (suggested in a previous article, which then should be quoted)
Or: It is suggested that… (the authors make this suggestion themselves in this article)
Author Response
Many thanks for your comments and advice on our manuscript. We have carefully revised the manuscript in response to you. And a native English speaker has checked and revised our manuscript. Changes in the revised manuscript are in red. Here are point-by-point responses:
1) Regarding the enzymatic activities tested, the authors should state that they also tested for lipase activity of the QS-proficient strains but did not detect any.
Reply: Many thanks for your advice. We have added the statement about lipase activity in lines 139-142 and lines 231-235.
2) What is the point of Fig S2? The conditions in which the activities are measured in this figure are not explained, and therefore it is difficult to compare the data shown there with those in Fig. 5. Thus, the phosphatase activity of P. suaedae is 1600 nMh-1 in one case and 140 nMh-1 (at the maximum) in the other. In addition, even if negative values are said to mean nothing, they are present in the figure, which should not be the case. I suggest removing Fig S2. It is not useful.
Reply: Thanks. We agreed with your comment about Fig. S2 and we have deleted the figure.
3)Please have your manuscript checked by native English speakers. Problems with the tenses remain. In the introduction the present tense should be used for general statements. For instance, it should be:Line 57: AHL-QS also plays an important role in regulating… Lines 320-321: hydrothermal vents have unique…. microbial populations live. This is also the case at many other places, which I decided was not my duty to point out.
Reply: Thanks. We have revised the tenses in sentences according to your comments. And a native English speaker has checked and revised our manuscript. Changes in the revised manuscript are in red.
4) Awkward/ incorrect sentences are found throughout the text. For instance:
Lines 352-354, it should be: …Pantoea ananatis B9, which belongs to Enterobacterales and was isolated from natural marine snow particles, could produce six AHLs, and its extracellular alkaline phosphatase activity …
Line 356, do the authors mean: It was suggested that … (suggested in a previous article, which then should be quoted)
Or: It is suggested that… (the authors make this suggestion themselves in this article)
Reply: Thanks. We have revised these sentences according to your comments (lines 284-286).